# Species Traits Drive Long-Term Population Trends of Common Breeding Birds in Northern Italy

**DOI:** 10.3390/ani11123426

**Published:** 2021-12-01

**Authors:** Pietro Tirozzi, Valerio Orioli, Olivia Dondina, Leila Kataoka, Luciano Bani

**Affiliations:** 1Department of Earth and Environmental Sciences, University of Milano-Bicocca, Piazza della Scienza 1, 20126 Milano, Italy; p.tirozzi@campus.unimib.it (P.T.); valerio.orioli@unimib.it (V.O.); olivia.dondina@unimib.it (O.D.); l.kataoka@campus.unimib.it (L.K.); 2World Biodiversity Association Onlus c/o NAT LAB Forte Inglese, Portoferraio, 57037 Livorno, Italy

**Keywords:** functional groups, life history, ecological traits, generalized additive models, population trend, birds

## Abstract

**Simple Summary:**

We assessed population trends for breeding birds in Lombardy (N Italy) from 1992 to 2019 and investigated the relationships between the observed trends and groups of species sharing similar characteristics (i.e., functional groups). We found a general positive or stable situation for 76% of the species. However, about 24% of the species declined, with worrying negative trends (greater than −50%) for two-thirds of them. Regarding species groups, we found that populations of migrants, of species with short incubation period, and of species with high annual fecundity declined. Similarly, populations of plant-eaters, of species feeding on invertebrates, and of farmland birds decreased. Only populations of woodland birds increased. In conclusion, our study provided a portrait of the status of common breeding birds in the region. Moreover, by analyzing the population response of the functional groups, we identified which of them experienced the most significant population changes, providing the foundations to implement studies aimed at quantifying the effects of specific divers responsible of the observed population changes in these groups.

**Abstract:**

Long-term population trends are considerable sources of information to set wildlife conservation priorities and to evaluate the performance of management actions. In addition, trends observed in functional groups (e.g., trophic guilds) can provide the foundation to test specific hypotheses about the drivers of the observed population dynamics. The aims of this study were to assess population trends of breeding birds in Lombardy (N Italy) from 1992 to 2019 and to explore the relationships between trends and species sharing similar ecological and life history traits. Trends were quantified and tested for significance by weighted linear regression models and using yearly population indices (median and 95% confidence interval) predicted through generalized additive models. Results showed that 45% of the species increased, 24% decreased, and 31% showed non-significant trends. Life history traits analyses revealed a general decrease of migrants, of species with short incubation period and of species with high annual fecundity. Ecological traits analyses showed that plant-eaters and species feeding on invertebrates, farmland birds, and ground-nesters declined, while woodland birds increased. Further studies should focus on investigation of the relationship between long-term trends and species traits at large spatial scales, and on quantifying the effects of specific drivers across multiple functional groups.

## 1. Introduction

Wildlife monitoring programs are fundamental to implement species conservation strategies and to verify to what extent they are effective [1,2,3]. Population trends represent one of the most informative proxy of the status of populations’ health, from local to continental spatial scales. One of the most monitored taxa worldwide are birds [4,5], an animal group of large conservation interest, target of several conservation policies in different countries, as well as at a continental scale (e.g., Birds Directive 2009/147/EC in European Union). From the past century, many bird monitoring programs have been started in many countries worldwide, both at a continental scale (e.g., the Pan-European Common Bird Monitoring Scheme—PECBMS—in Europe [6], the North American Breeding Bird Survey—BBS—[7]) and at a national scale (e.g., in Finland [8,9], the Breeding Bird Survey—BBS—in UK [10]). In other cases, many data is available, although these have often been collected within projects with a specific purpose other than long-term monitoring [11]. Historical data collected in different years by different projects are surely a valuable but often overlooked source of information, available for the evaluation of the conservation status of species, communities, and ecosystems [12,13,14,15]. Beyond the assessment of population dynamic of each bird species, which is essential to define species-specific conservation priorities and actions, birds represent valuable ecological indicators that can potentially provide useful information about the overall biodiversity and ecosystem processes [16,17,18,19,20]. Changes in species assemblages can be driven by species-specific responses to environmental perturbations (either natural or anthropic), and identifying which factors cause the observed changes is crucial to mitigate the impact of these perturbations. Although species population trends are not directly informative on drivers of population changes [21], observing a common trend among species with similar ecological or life history traits can provide the foundation to test specific hypotheses about the drivers of the observed population dynamics [22]. Despite the arising attention of a trait-based approach in community ecology [15,23,24,25,26,27,28,29] and population ecology [30,31,32,33,34], only in the last few years researchers have started to explore the link between traits and population trends in avifauna over long time periods, highlighting the existence of different patterns among distinct functional groups [22,35,36,37,38,39,40,41].

In Europe, the decline of long-distance migrants wintering in Sub-Saharan Africa compared to short-distance migrants or sedentary species has been emphasized in several studies (e.g., ref. [22,35,42]), as well as the general decline of farmland birds, as underlined by the tendency of the Common Farmland Birds Indicator [6]. Gregory et al. [43] highlighted the existence, in the period from 1980 to 2015 and across 28 European countries, of different trends among farmland species and generalist forest species, with the first ones decreasing and the second ones stable. However, some contrasting trends were found for old forest species due to forest management [9]. Insectivorous birds are declining in Europe [37]. In Germany, however, this trend has not been confirmed for the entire guild, since differences emerged between trends of farmland insectivorous birds and forest insectivorous birds, with the former decreasing and the latter stable [22]. Moreover, species with narrow niche breadth (i.e., specialist species) may suffer more from human-driven environmental alterations than species with broad niche breadth (i.e., generalist species), resulting in contrasting trends between the two groups [40,44]. Nesting location can influence the breeding success by affecting the exposure of the nest to predation (e.g., birds or mammals, [45]). For example, ground-nesting bird populations have decreased during recent decades in Europe [46,47] with most dramatic declines reported for larger species such as waders and bustards inhabiting agricultural landscape [48,49] or forest grouses (Tetraonidae, [50,51]). In addition, many others less explored life history traits (e.g., incubation and fledging period, body mass, fecundity, species’ mobility) can provide useful insights about the relationship between population trends and groups of species sharing similar traits.

In Italy, where a long-term monitoring program at national scale is available only from 2000 (MITO2000 projects, [52,53]), no studies explored this relationship over long time series. However, in Lombardy (a region in the north of Italy), bird data from different projects have been collected for almost 30 years (1992-ongoing), and they can be merged to assess long-term population trends in the region, while accounting for different sources of data heterogeneity (i.e., environmental bias, overdispersion and zero-inflation, [54], see Section 2.3). In this study, we aimed at estimating trends for common breeding birds in Lombardy from 1992 to 2019, and at investigating how species traits are linked to population trends, in order to identify which species need of major conservation attention and which are the functional groups that are experiencing the most relevant demographic changes.

## 2. Materials and Methods

### 2.1. Study Area

The study was carried out in Lombardy, a region of 23,861 km^2^ in Northern Italy (45° N, 9° E). The northern part of the area is characterized by mountains Alps and Prealps, separated from each other by the wide glacial valley of Valtellina. The southwestern corner is characterized by the hilly landscape of northern Apennines. The rest of the region hosts a large portion of the alluvial plain of the Po River (the widest plain in Italy) extending from west to east for more than 12,000 km^2^. The average elevation of the whole region is 610 m above sea level (a.s.l.), but it shows a large variation across different areas, ranging from 2 m a.s.l. of the plain to roughly 4000 m a.s.l. of the Alps. Land use is characterized by urban areas (14.7%), agricultural lands (42.2%), and natural and semi-natural lands (39.6%, of which 61.4% are forests). Alps and Prealps are mainly characterized by coniferous forests, meadows and grasslands at higher elevations, and deciduous forest at lower ones. The Apennine area is characterized by vineyards, extensive farming and deciduous and mixed forests. The Po Plain is heavily men-modified, with intensive cereal cultivations (mainly maize) in the central and eastern part, and dense urban areas and paddy fields in the west (Figure 1).

### 2.2. Survey Design and Bird Data

Data were collected from five different projects on breeding bird surveys in Lombardy carried out from 1992 to 2019. Overall, the dataset consists of 18,505 point counts, with an average of 771 point counts per year (range 373–1443; SD = 244). No data are available for years 1993, 1994, 1997, and 1998 (Appendix A). The first project was the “Long-term Monitoring Program”, launched in 1992 and covering 18 years. It consisted in a stratified sampling design, where seven primary sampling units were defined based on the main landscapes present in the Region. Within each primary sampling units, secondary sampling units, corresponding to about 10 km × 10 km squares (“Tavolette IGMI”, 1:25,000 maps) were extracted proportionally to the representativeness of each landscape, and yearly renewed to guarantee complete coverage of the Region in the long-term. Point counts were randomly located within secondary sampling units, according to territory accessibility. From 2007 to 2016, fixed secondary sampling units (20 in 2007, 21 in 2008, 23 in 2009–2010, 24 in 2011–2016) were added to the random ones. Finally, starting from 2017, all sampling units became fixed (34 units per year, including all of them previously performed). The “Regional Fauna Database” project was carried out from 2000 to 2006, using a systematic stratified sampling design. Finally, other three projects (“Forest Biodiversity Survey”, “Lowland Biodiversity Survey”, and “Greenway Project”) were performed in restricted sub-areas of the region during a limited period (few years) with the aims of surveying breeding birds living in forests, in agricultural lands and in the Apennines, respectively (Appendix A). In all projects, data were collected using a single-visit point-count method with unlimited distance [56]. The minimum distance between sampling locations was 500 m within each year. All birds heard or seen in 10 min were recorded [57]. Bird surveys were performed during the breeding season (10 May to 20 June) to minimize the count of migrants (birds not breeding in the study area) and to survey territorial birds. Surveys were conducted from sunrise to 11.00 a.m., only in good weather conditions, sunny to cloudy, without rain or strong wind [58]. This technique provides a measure of relative bird abundance [59,60]. The technique is effective in detecting bird species belonging to the orders Columbiformes, Cuculiformes, Apodiformes, Coraciiformes, Piciformes, and Passeriformes [58], but can also be used to survey some other common species, such as the Common Buzzard (*Buteo buteo*) and the Common Kestrel (*Falco tinnunculus*). All counts were expressed in number of breeding pairs according to Blondel et al. [56]. For early-breeding sedentary species (e.g., tits) that in our study area might start to breed in February-March, the sampling period might not entirely fit. Actually, this may represent a weakness in the detection of such species. However, since the early-breeding sedentary species can be detected in groups composed by parents and offspring during the late spring, each family group was converted into one breeding pairs in order to not overestimate the counts. Although some limitations can arise for some species starting to breed in the early spring (e.g., woodpeckers), the survey window we set (10 May to 20 June) can be considered suitable for most of species that can be detected by point counts technique in our study area. Additionally, even if an underestimate of detected individuals whose territorially behaviour peak precedes the beginning of the survey period, this source of bias remain constant over years. This way, counts of early-breeding sedentary species do not affect the trend estimates. In the case of large-scale monitoring programs, it is essential to obtain an optimal trade-off between costs (i.e., within-season multiple surveys effective to detect early- and late-breeders) and benefits (i.e., large representative sample). Thus, we prefer to obtain a large representative sample, relying on single-visit survey, accepting an underestimation of early-breeders that do not jeopardize the trend estimates.

Since data did not rely on within-season multiple surveys, it was impossible to account for species detection probability. However, although the detection probability might represent an issue to be addressed in trend analysis relying on density estimates [61], while using relative abundance data coming from a large dataset, where the effect of stochasticity on species detection is limited [25], accounting for the detection probability can be considered superfluous (Appendix B). Among the 204 species detected in the whole period, we modelled trends for those with an overall relative frequency higher than 2%.

To avoid an overestimation of the number of breeding pairs for gregarious species during the breeding season, whose groups are composed of parental pairs and fledglings, we used the following conversion factor (CF) (i.e., the number of individuals considered as one breeding pair [62]): Feral Pigeon (*Columba livia domestica*), CF = 8; Long-tailed Tit (*Aegithalos caudatus*), CF = 14; Common Starling (*Sturnus vulgaris*), CF = 11; Italian Sparrow (*Passer italiae*) and House Sparrow (*Passer domesticus*), CF = 14; Eurasian Tree Sparrow (*Passer montanus*), CF = 16. For instance, in the case of the Common Starling, the detection from one to 11 individuals has been converted into one breeding pairs, from 12 to 22 individuals into two breeding pairs, and so on.

### 2.3. Species Trend Assessment

Our dataset is affected by environmental bias inherited from spatial bias (i.e., the inadequate representation of the variability of environmental covariates in the study area [63,64]), due to differences in the sampling design of the five projects merged to obtain the final dataset. This issue prevents the trend assessment by using original data [54]. Furthermore, overdispersion (i.e., the variance is larger than the mean) and zero-inflation (i.e., a particular type of overdispersion due to an excess of zero counts) could be present in count data [65,66,67], and ignoring them can lead to serious errors in the interpretation of results from an ecological perspective [68]. To handle environmental bias, overdispersion and zero-inflation, we adopted the modelling procedure described in [54], which is summarized in the following part of the Section and to which the reader can refer for a more detailed explanation. In their work, authors assessed long-term population trends for six bird species adopting a full-factorial design to account for environmental bias, overdispersion and zero-inflation. Since environmental bias resulted the most relevant factor to determine the magnitude of trend estimates, in our analyses we excluded models not accounting for that. Following the aforementioned approach, we performed, starting from the original count data, four generalized additive models (GAMs [69,70,71]), namely C-ZIP-GAM, C-ZINB-GAM, C-P-GAM, C-NB-GAM (C, model with covariates dealing with environmental bias; ZIP, zero-inflated Poisson; ZINB, zero-inflated negative binomial; P, Poisson; NB, negative binomial). GAMs allow for relaxing parametric assumptions of generalised linear models (GLMs [72]), replacing some, or all, of the parametric terms by smooth functions of the covariates. We used the year of survey as a parametric factor; 17 land cover variables (continue urban matrix and infrastructures, discontinue urban matrix, arable lands, paddy fields, vineyards, orchards, olive groves, wood plantations, meadows and pastures, broadleaved forests, mixed forests, coniferous forests, grasslands, shrub lands, areas with sparse or absent vegetation, wetland vegetation, rivers and streams) recorded as percentage cover within a 250 m circular buffer around each survey site and four topographic variables (average values within a 250 m circular buffer around each survey site of elevation, sine and cosine of the aspect, slope) as covariates to account for environmental bias; and the spatial trend (interactions between X and Y coordinates (UTM 32N, WGS84 Datum)) to account for spatial pattern in the data [73] and potential spatial autocorrelation [12,25]. For the zero-inflated GAMs, elevation and the percentage cover of urban and forest area within a 2500 m circular buffer designed around the survey site were used as predictors to explain the zero-inflation process. All covariates were modelled as smooth function (maximum degree of freedom set at four), except for the sine and cosine of the aspect which were modelled as linear effect. Based on the AIC [74], for each species, the best of the four models was picked out. Thus, it was used to predict yearly population indices through a parametric bootstrap with 1000 simulations [75]. We used the median of the distribution of bootstrapped predictions as population index estimator for each year, and calculated the 95% confidence intervals by the percentile method [76]. Land cover values in the prediction matrix were predicted from GAMs wherein the yearly average values for each covariate, derived from the six-digital land-use vector map DUSAF available from 1980 to 2019 (DUSAF 1980, 1999, 2007, 2012, 2015, and 2018, downloadable from http://www.geoportale.regione.lombardia.it/, accessed on 21 September 2021) were fitted as smooth function of year. In addition, since topographic variables and coordinates are time-invariant, they were fixed at the overall mean and the centroid of the area, respectively. Since some species could have clustered or geographically restricted distribution (e.g., alpine species), we used point counts in which the species was present in the period 1992–2019 to determine the species distribution area (km^2^) across the region (Minimum Convex Hull method, or MCH). For species that had a ratio between the MCH and the regional area less than 0.80, we used the MCH as polygon to extract values for all covariates; otherwise, the entire Region was used.

To assess the long-term population trends, a weighted least square linear regression (WLS) was performed, using the median of the yearly population index as response variable (estimated number of breeding pairs per sampling site), the year as continuous explanatory variable, and the reciprocal of the width of the confidence interval associated to the yearly estimate as weight. We acknowledge that yearly indices are not temporally independent, but since our aim was to assess long-term trends, a linear regression can be considered adequate for the purpose [77,78]. The model can be summarized as $I_{t}=\beta \times Year+\varepsilon_{i},\;w=1/CI$, whereby we tested whether the trend was significantly different from zero (p(β) ≤ 0.05). The variation in population dimension from 1992 to 2019 (T%) was quantified as $T\%=\left[\left(I_{2019}-I_{1992}\right)/I_{1992}\right]\times 100$.

Geospatial analyses were performed in ESRI ArcMap 10.7.1 (Redlands, CA, USA), while trend analyses in R software [79] using the packages *mcgv* [80] and *zigam* [81]. Since these last analyses were particularly resource-consuming, they were performed on the supercomputer Marconi-A3 HPC [82].

### 2.4. Trait-Based Analysis

#### 2.4.1. Bird Traits

To explore whether species sharing similar life history and ecological traits showed a similar demographic signal in response to specific drivers, we selected a set of 12 traits (Appendix A). After evaluating the association among each pair of traits (categorical variables) through the Cramer’s V coefficient [83] using R package *vcd* [84]), we retained those traits not significantly associated with the others (Chi-Square test of independence or Fisher’s exact test for expected frequencies less than five). In case of significant relationships, we retained only traits with a weak or moderate degree of association (Cramer’s V < 0.5 [85,86]; Appendix A). After this check, four life history and four ecological traits were selected for the following analyses. Life history traits encompassed the migration strategy, dispersal ratio, annual fecundity, and incubation period. Migration strategy was derived from [87], which reports detailed information about phenology of populations in the study area. We classified species into three groups: “sedentary species” (non-migrants), “short-distance migrants” (wintering in Europe or North Africa), and “long-distance migrants” (wintering in Sub-Saharan Africa). Dispersal ratio (mean wing length [mm]/cube root of mean body mass [g]) was used as index of species’ mobility [88]. Annual fecundity was calculated as the product of average clutch size and average number of broods per year [89], and incubation period was the mean duration of eggs’ incubation (days). We derived these information from [90]. For the Common Cuckoo (*Cuculus canorus*), we set to one the number of average broods per year. We transformed dispersal ratio, annual fecundity and incubation period from numerical to categorical variables, following the approach suggested by [22]. We used the first quartile of the variable values of all species considered in the analyses (Dispersal ratio: values ≤ 28.278, Annual fecundity: values ≤ 5, Incubation period: values ≤ 13) to identify the groups “low dispersal ratio”, “low annual fecundity”, and “short incubation period”. The upper quartile (Dispersal ratio: values ≥ 33.737, Annual fecundity: values ≥ 10.625, Incubation period: values ≥ 17.25) was used to define the levels “high dispersal ratio”, “high annual fecundity”, and “long incubation period”. Finally, the values in-between were considered to represent species with intermediate characteristics. Ecological traits included diet, nest type, landscape type, and degree of specialization. Diet represents the main source of food on which the species feeds on. It was derived from information collected in [91], where the percentage of food items used by the species is reported. We assigned species to the level “vertebrates” if it feeds on at least 70% of vertebrates, to “plant-eaters” if it feeds on at least 70% of vegetal material (seeds, nectar, fruits, other part of plants), and to “invertebrates” if it eats at least 70% of invertebrates (arthropods, mollusks, annelids). We assigned species to “omnivores” whether none of the foregoing categories individually exceeded the threshold of 70%. The Black Kite (*Milvus migrans*), resulting as scavenger for at least 70%, was classified as “vertebrates”. Nest type represents the nest position and it was derived from [90]; we reclassified the original categories “open-arboreal” and “closed-arboreal” into the group “elevated-nesters”, “ground” and “ground-closer” into “ground-nesters”, and finally “hole-nesters” were retained as in the original study. Common Cuckoo was assigned to “elevated-nesters” [92]. Landscape type represents the habitat preference of the species at landscape scale. To derive this trait we used our dataset that, being of large dimension and temporal coverage, can be considered a reliable source of information on habitat usage by the species. We calculated the median of fractional cover for each level (urban, agricultural, forest, natural open-habitat, wetlands-rivers-lakes, classified according to DUSAF map [55]) within a radius of 2500 m around the point counts where the species was present. Point counts of each year were related to the temporally closest available DUSAF or CORINE [93] digital map. Whether the median of the fractional cover of a specific level was greater than 50% we assigned that level as landscape type, namely “farmland”, “woodland”, and “natural open-habitat” (such as shrubs, grasslands, rocks). Whether the median did not exceed 50% for any levels we assigned the category “several”. To calculate the degree of specialization we started from information collected in [92]. We calculated single-species specialization indices [94] for each of the five following traits: food type, acquisition behavior, substrate from which food is acquired, foraging habitats, nesting habitats. Subsequently, we derived an overall specialization index (SI) by computing the mean of the five indices. Hence, the higher the SI, the greater the species specialization. In foraging habitats, we added “urban” and “garden” as habitats used by the Feral Pigeon, and “dry grassland”, “urban” and “garden” by the Eurasian Magpie (*Pica pica*). Moreover, in the substrate from which food is acquired category, we grouped the levels “watersurface”, “underwater” and “water” into a single level.

#### 2.4.2. Relationship between Population Indices and Traits

To test whether species belonging to the same functional group showed similar responses in terms of population trends, and if differences among trends of different groups were significant, we used a distinct weighted linear regression model for each trait. The median of the yearly population index was considered as response variable, “year”, the trait and their interaction as explanatory variables, and the reciprocal of the width of the confidence interval associated to each median yearly index was used as weight. The interaction term represents the trend for the specific group. In order to test whether the coefficient for each interaction term was significantly different from zero and from the other levels, we carried out additional statistic tests using the finite-sample F statistic by the function *linearHypothesis* of the R package *car* [95].

As the absolute value of the population index is not directly comparable among different species since it depends on the regional species abundance, we divided, within each species, the median of the yearly index by the median value of the index in the first year of the time series (i.e., I_1992_). Exceptions were made for the Great Cormorant (*Phalacrocorax carbo*) (I_2005_), Northern Lapwing (*Vanellus vanellus*), Short-toed Treecreeper (*Certhia brachydactyla*) and Common Linnet (*Linaria cannabina*) (I_1995_) since they were not recorded in 1992. To obtain comparable confidence intervals (used as weights in the model), we adopted the identical procedure for the 2.5th and the 97.5th percentiles. In this way, we assigned the value one at the median population index in the first year, and all the others indices (median, 2.5th and 97.5th percentiles) were expressed as variation in relation to the median of the first year.

## 3. Results

### 3.1. Species Population Trends

We assessed population indices and trends for 76 breeding bird species. According to AIC ranking, C-P-GAM (Poisson GAM with covariates dealing with environmental bias) was picked out as best model for 12 species, C-NB-GAM (negative binomial GAM with covariates dealing with environmental bias) for nine, C-ZIP-GAM (zero-inflated Poisson GAM with covariates dealing with environmental bias) for 40, and C-ZINB-GAM—zero-inflated negative binomial GAM with covariates dealing with environmental bias—for 15 (Appendix A). In the selected models (see Appendix A for models’ summary), the explained deviance ranged between 10.90% and 77.80% (mean = 38.61%) in the non-zero-inflated GAMs, while it ranged between 52.90% and 97.20% (mean = 82.62%) and between 7.30% and 66.60% (mean = 30.27%) in the binomial (i.e., zero-inflated part) and count component of the zero-inflated GAMs, respectively (Appendix A). Long-term trend analyses resulting from the weighted least square linear regression (WLS), highlighted a significant positive trend (T%) for 34 species (ranging from + 12,060% for the Great Cormorant to + 14.45% for the Eurasian Blackcap), negative for 18 species, 12 of which experienced a population decrease larger than 50%, while 24 species did not show a significant trend (*p* value > 0.05; Table 1 and Appendix A).

### 3.2. Relation between Population Trends and Species Traits

#### 3.2.1. Life History Traits

Results for life-history traits were summarized in Table 2A–D. Sedentary species showed an increasing but not significant trend, while short and long-distance migrants significantly decreased (Figure 2A), but the magnitude of decline did not differ significantly between these two groups (Table 2A, *p* value = 0.517). Species with intermediate dispersal ratio showed a significant decreasing trend (Figure 2B), even if it was not significantly different from the trend estimated for species with high and low dispersal ratio (Table 2B, *p* value = 0.194 and *p* value = 0.295, respectively). Analyses for the annual fecundity showed that species with high annual fecundity decreased (Figure 2C), and their decline was significantly different from the (negative but non-significant) trends of the other two groups (Table 2C). Finally, concerning the incubation period, only species with low incubation period showed a significant trend, which was negative (Table 2D and Figure 2D).

#### 3.2.2. Ecological Traits

Results for ecological traits were summarized in Table 3A–D. As regards the diet, pairwise comparison among groups did not show significant differences. However, tests for a non-zero slope within each group revealed a significant decline for plant-eaters and species feeding on invertebrates, while omnivores and species eating vertebrates did not exhibit any significant trend (Table 3A and Figure 3A). Ground-nesters showed a decreasing trend, while neither elevated-nesters nor hole-nesters resulted to have significant population changes (Table 3B and Figure 3B). Species inhabiting farmland landscapes showed a clear decrease, while species preferring woodland landscapes resulted to be on increase (Figure 3C). Although species of natural open-habitat showed a raising tendency, the trend was not significant, maybe due to the limited sample dimension for this group (nine species). Finally, for the group of species classified as “several” (i.e., those not having a prevalent habitat preference) the regression coefficient for the interaction term was roughly equal to zero (Table 3C). In relation to the overall specialization index, the model highlighted a significant negative trend for species with a medium degree of specialization, while both specialists and generalists did not show significant population changes over the period (Figure 3D). However, no differences were detected among trends of the three groups, very probably due to the weak dissimilarity among the coefficients (Table 3D).

## 4. Discussion

### 4.1. Modelling Approach

The use of adequate models to obtain reliable estimates of yearly population indices is fundamental to assess non-distorted population trends, whose evaluation is of extreme importance in wildlife conservation. In this context, our results showed that zero-inflated models should be used more than they actually are. Specifically, for 72% of the 76 species analyzed, zero-inflated GAMs resulted to be the most suitable models for predicting the yearly population indices. Unexpectedly, zero-inflated models are often overlooked in population trend analysis without an objective reason, but their use should be always tested in case of a high number of zero counts in the data. Overdispersion, which is often detected in count data deriving from multispecies surveys [58,97], can actually disappear or decrease if zero-inflated models are used (Appendix A). The management of zero counts and the error structure for the response variable can strongly affect trend estimations [54], and it should not be carried out a priori but after a modelling selection procedure, which compares multiple candidate models.

### 4.2. Species Population Trends

The long-term trend analysis allowed obtaining an updated portrait of the conservation status of 76 common breeding birds in Lombardy. Although roughly three-quarters of them showed non-significant or increasing population trends, 24% of the studied species resulted on decline. The Eurasian Skylark (*Alauda arvensis*) showed a worrying trend: its population was reduced by 99.65% since 1992, pointing out the extremely critical situation that the breeding populations of the species are experiencing in the Region. In a previous work, Bani et al. [58] found a reduction of 75% of the species regional population for the period from 1992 to 2009; after 2009, the population indices estimated by our analyses revealed a further decrease until 2015 (Appendix A), resulting in a stronger population decline, overall. However, at national and continental scale, the species exhibits a minor accentuated decrease [6,98]. In farmland, this species is very sensitive to the intensive agricultural management (with the use of pesticides, fertilizers, and excessive mowing) as well as to the decrease in crop diversity, both leading to unsuitable conditions for nesting [99,100,101,102]. Urbanization around arable land negatively affect the occurrence of the species [103], as well as the anthropic alterations in natural and semi-natural grassland in mountains areas [104]. In the Alpine area of Lombardy, the distribution of the Eurasian Skylark might be adversely affected by shrubs and forest expansion, as shown for other alpine species [12], but this pattern needs to be confirmed. Other five species (the African Stonechat *Saxicola torquatus*, the European Goldfinch *Carduelis carduelis*, the European Greenfinch *Chloris chloris*, the Eurasian Wryneck *Jynx torquilla*, and the Red-backed Shrike *Lanius collurio*) showed very strong negative trend, with population declines greater than 80%. The African Stonechat experienced a sharp drop from 2006 (Appendix A). The decline resulted to be greater than reported at national scale in Italy, where a decrease of 68.5% was detected from 2000 to 2020 [98]. However, in the close Switzerland, a region environmentally and climatically similar to the Alpine area of Lombardy, the species appeared to be on increase [105], and in Europe it seemed to be stable [6]. These contrasting trends can derive from local factors (e.g., land management) that, acting at smaller spatial scale, can produce different effects in breeding areas [32]. For this species, land consolidation of arable fields, leading to the degradation of rural landscape structure, might act as ecological traps [106]. Trends for the European Goldfinch and the European Greenfinch showed a more drastic situation compared to the known status at national scale in the last twenty years [98]. The decline found by Bani et al. [58] in Lombardy seems to be continued over the last decade. However, in Europe these two finches did not decline over long period [6], or showed fluctuant dynamics, suggesting the existence of different drivers, such as seeds availability or epidemics, acting at local spatial scale [107]. In addition, both the Eurasian Wryneck and the Red-backed Shrike highlighted a more marked decline than that shown at national [98] and European [6] level. In the study area, the two species mainly inhabit semi-open environments, especially extensively managed farmland. Agricultural intensification, leading to loss of suitable foraging and nesting sites, as well as to a reduction of preys (mainly represented by insects) availability and detectability, may have played a key role in their decline [106,108,109,110,111]. We noted that other five species, the Barn Swallow (*Hirundo rustica*), the Western Yellow Wagtail (*Motacilla flava*), the Italian Sparrow, the Cetti’s Warbler (*Cettia cetti*), and the Common Swift (*Apus apus*) experienced a significant decline ranging from 50% to 80%, higher than the known situation at national scale [98,112]. The Barn Swallow, in our study area, showed a population decline of rough 50% during 1990s and 2000s [58,113,114], and the negative trend continued over the last decade. Causes can be linked to both changes in agricultural practices as well as cattle farming [115]. The Western Yellow Wagtail, which in Lombardy is associated to farmlands, may have suffered from the intensification and changes in agricultural practices, in particular from loss of late mown areas [116]. The Italian Sparrow, in agricultural breeding areas, could be negatively affected by the progressive shift of cereal cultivations from wheat to maize, other than changes in livestock farming practices [117]. In urban settlements, the food shortage (e.g., caused by a more efficient street cleaning and by a reduction in “weedy” areas), the reduction of suitable breeding sites, predation and competition had likely contributed to the species decline [117,118,119]. Both the Common Swift and the declining Common House Martin (−45.55%) might have suffered from climatic factors acting in wintering and breeding grounds, but also along migration flyways [120,121]. Moreover, such species living in urban areas may suffer from different kinds of pollutants, as shown by Miniero et al. [122]. Less clear remain the drivers of the negative long-term trend of the Black Kite (−54.96%), although some study highlighted the possible effect of electrocution and road casualty [123]. As for many other long-migrants species, we do not exclude that other factors might act in sub-Saharan wintering areas (see Section 4.2).

Among the species resulting on increase, the Great Cormorant exhibited a remarkable positive trend (+12,060 %) in the last fourteen years. This finding suggest the need of further and specific investigations in order to assess in detail the absolute dimension of the population, considering the impact that the species may have on herons [124], fish species [125] and economic activities as fisheries [126]. All tits (included the Long-tail Tit), resulted significantly on increase, except the Coal Tit (*Periparus ater*) that showed a positive but non-significant trend. These findings are largely in accordance with trends observed in Switzerland during the same period [105], while in Europe, during the last forty years, the Marsh Tit (*Poecile palustris*), the Willow Tit (*Poecile montanus*) and the European Crested Tit (*Lophophanes cristatus*) declined [6]. Also in Finland, these last two species declined due to intensive forestry [127], and the Willow Tit has been added in the National Red List [128]. We found a considerable increase of the population of the Northern Lapwing (+1653%), a result quite in contrast with the negative trend observed in Europe [6], especially in arable land [129,130], which is the type of habitat where the species mainly occurs in Lombardy. However, the first case of breeding of the Northern Lapwing in our study area is reported in 1960s, and from 1980s to the first decade of the twenty-first century the population has grown from a few hundreds of individuals to about two thousands of breeding pairs, probably supported by the water management of paddy fields, which represent the favorite breeding habitat for the species in Lombardy [131,132,133]. However, changes in agricultural practices for this kind of crops (in particular the delay of the flooding period) occurred in the last years, can negatively affect the reproductive success of the species, pointing out the importance to carry on specific demographic studies. Eurasian Magpie showed a strong increasing trend (+754%). It resulted greater than the average increment at national scale [98], and in contrast with the European situation, where the species from the early 1990s exhibited a clear decrease [6]. In our region, where the urbanized areas increased during the last decades [55], the positive tendency of the Eurasian Magpie can be linked to its capacity of adapting to highly urbanized habitats. In this type of environment, the density of pairs can reach higher values than in non-urbanized habitats [134], maybe favored by high food availability. We have not evidence that the species increase is linked to a reduction of human persecution, as occurred in other areas (e.g., [135]). Song Thrush (*Turdus philomelos*), Common Wood Pigeon (*Columba palumbus*), Great Spotted Woodpecker (*Dendrocopos major*) and Short-toed Treecreeper showed strong positive trends, with regional population increased tenfold or more from the beginning of the time series. These findings confirmed the trends of the species in Europe, where from the early 1990s remarkable increments of the continental population were detected [6].

Population trends are one of the most important criteria for the conservation status assessment of species (e.g., [136]). Looking at the IUCN Italian Red List [137], we found that six of the 18 declining species are listed as “vulnerable”, and one of them, the Eurasian Wryneck, as “engendered”. The European Goldfinch and the European Greenfinch, which showed strong decreasing trends, are listed as “near threatened”. Although our results are at regional scale, they can provide valuable information for updating local, national and continental Red Lists [39].

### 4.3. Relation between Trends and Species Traits

The trait-based approach proved to be very useful to infer on general drivers acting on wildlife populations, laying the foundations for more specific studies with the aim of identifying the proximal causes shaping bird population dynamic. Our analyses detected a general decline of migrant species, and a positive but non-significant trend for sedentary species. The decline of long-distance migrants has been extensively outlined for European breeding birds [42,138,139], with some exceptions (e.g., in Estonia [140]), and several explanations have been proposed (e.g., drought of Sahel areas, habitat loss and degradation, hunting, phenological mismatch [42,138,141,142,143,144,145,146]). Moreover, long-distance migrants were proved to decline faster than short-distance migrants in Europe [22,35,42], but we did not detect such a difference between the two groups, corroborating the idea that geographic variation in trends may occur [139].

The link between dispersal ratio, as surrogate of species’ mobility, and the population trend did not highlighted a clear gradient for this trait. In a previous study, Crawford et al. [147] found that a lesser species’ mobility was associated with a high rate of adult mortality. Although our results did not confirm this finding, and differences among the three groups did not emerge, a signal that long-term population dynamic can vary in relation to species’ mobility was found for species with intermediate mobility. Further studies should focus on this life-history trait that is very poorly investigated in trend analysis.

Species with high annual fecundity exhibited a clear negative trend. This finding was in accordance with results underlined by Jiguet et al. [89] for common breeding birds in France. Annual fecundity strongly influences variations in population size in short-lived species [148], and it is possible that individuals of such species will have too short a lifespan to learn and adapt to directional environmental changes. Moreover, density-dependent effects might also explain the observed pattern: high fecundity could lead to an increase in juvenile mortality, due to increased intraspecific competition [89,149].

Results for the incubation period highlighted that species with shorter incubation period decline, while those with longer incubation period showed a positive tendency, although it was not significant. On one hand, species with longer incubation period might be longer exposed to anthropogenic and ecological pressures during a critical stage of their life cycle. However, Liu et al. [150] highlighted that the incubation period is strongly and positively associated with nest concealment. Species that are able to hide their nests more efficiently, may be less exposed to predation risk, with positive effects on fitness in the long period. Additionally, shorter incubation periods are often found in species that face higher levels of nest predation [151,152]. However, predation risk may also be a habitat-dependent process [153]. Moreover, human activities can contribute to enhance this risk, as well as they can directly affect hatching success [154,155]. Further studies to investigate the effects of anthropogenic pressures on hatching success across different types of habitat can provide novel insights for planning ad hoc conservation strategies. 

Analyses of ecological traits showed a decline of plant-eaters and species feeding on invertebrates. The decrease in insect populations has been largely recognized [156] and reported in several countries in Europe (e.g., [157,158]), as well as worldwide [159], but can only partially explain the general decrease of insectivore birds in Europe in the period 1990–2015 [37]. In our study, several species that eat invertebrates, such as the Northern Lapwing, the European Bee-eater (*Merops apiaster*), the Short-toed Treecreeper, the European Green Woodpecker (*Picus viridis*), the Spotted Flycatcher (*Muscicapa striata*), the Common Firecrest (*Regulus ignicapilla*) and the Melodius Warbler (*Hippolais polyglotta*) exhibited strong increasing trends (>300%), highlighting that other component (e.g., habitat preferences) can interact with food habits in shaping population dynamic [22,37,160]. The decline of plant-eaters species (in our work mainly represented by species feeding on seeds) emerged in our study was in accordance with results in Europe [37]. In particular, plant-eaters linked with urban settlements and farmland, such as the Italian Sparrow, the European Goldfinch and the European Greenfinch, showed stronger negative trends. The use of herbicide and the disappearance of spring-sown cereals at the cost of winter-sown grains represent one of the major threats for the guild in farmland [161]. Additionally, the use of pesticides can have sub-lethal effects, such as impairing migratory ability in seed-eating species [162].

As regards to nest type, the general decrease of ground-nesters (largely farmland birds) can be associated with their higher predation risk [45,150,163], which might also be linked to agricultural practices that reduce the concealment of nests by modifying the habitat structure [164].

Farmland species showed a strong decline (roughly equal to that of short-distance migrants, see Table 2A and Table 3C for comparison between regression coefficients). This finding is not surprising, since the decrease of farmland avifauna in Europe is well known [6,43], as well as documented in Italy in the last 20 years (with a stronger negative trend in Lombardy compared to the situation at national scale [98]). Causes of this decline are ascribed to agricultural intensification [161], which lead to simplification and homogenization of agricultural landscapes (land consolidation, loss of fallows), reduction of crop diversity and massive use of pesticides. Moreover, changes in agricultural management, such as increasing number of cuts, dense swards and reseeding with high-yielding grass crops, can lead to an increase of human disturbance and to a decrease of habitat suitability for many species. It is probably that all or a part of these processes have played, and still play, an important role in our study area, where intensive agriculture is widely spread. For example, in Lombardy the average harvest for maize and autumn-sown grains per hectare, thought to be a good indicator of agricultural intensity [165], was 9.3 ton/ha in the period 2006–2019, while the national average in the same period was 5.3 ton/ha [166]. However, 13 of the 31 farmland species showed significant positive trends, highlighting that species-specific drivers can occur and determine different demographic responses. For instance, some thermophilic farmland species, such as the European Bee-eater, showed strong positive trends, probably favored by temperature increasing [167,168,169]. Woodland birds, which appear stable or increasing in Europe [22,43,170], confirmed this pattern in our study area. The increase in wood extension occurred in Lombardy over the last 40 years [55], both due to natural processes linked to the abandonment of mountain meadows and pastures and to human-made reforestations of bare grounds, may have increased the amount of suitable and available habitats for this group of species. This can have led to the establishment of new sub-populations and to the reduction of intraspecific and interspecific competition. Additionally, it is not to exclude that possible changes in forest management, resulting in an increase of forest quality, may have favored the positive trend of forest birds [171,172,173,174], but see [9]. Among woodland species, only two of them showed a negative trend: the Common Chiffchaff (*Phylloscopus collybita*), with a population reduction of 40.64%, and the Goldcrest (*Regulus regulus*), with a decrease of 41.42%. The Common Chiffchaff was also detected on decline by Bani et al. [58], but its trend is quite different compared to the observed trends in Europe [6], in Italy [112] from 2000 to 2014 and in other European countries (e.g., Germany [22], Switzerland [105], Sweden [175]). The species has many traits (long-distance migrants, intermediate dispersal ratio, ground-nesters, high annual fecundity, feeds on invertebrates) linked to negative trends, and it is probably that one or more of these species-specific characteristics are responsible of the observed negative trend in our study area. The Goldcrest experienced a general decline in Europe [6] as well as in Italy [112]. It is a cold adapted and forest specialist species, and the combination of these two factors may be the cause of the decline [43,176]. Species of natural-open habitat, which represent nine of the 11 species living in high mountain areas, did not show a significant trend. The finding is in accordance with that reported in some European mountains during the period 2002–2014 (i.e., Alps, Apennines), although in other mountain areas, such as Fennoscandia and Iberia, bird species declined significantly [177,178]. Indeed, mountain birds can differently respond to ecological drivers (e.g., climate change and land use changes), affecting local population trends and distribution [12,178].

Finally, we did not find a strong relationship between trends and the overall specialization index. This result did not fully support findings highlighted in previous studies. For example, Morelli et al. [40] underlined a negative relationship between the degree of specialization and population trends for 139 species in Europe, in particular in relation to nesting site specialization. Similarly, Kamp et al. [22] found a decline of specialist birds in relation to habitat breadth and diet breadth (note that categorization of these traits differs from the approach adopted in this study). It is clear that bird communities are moving to assemblages composed by more generalist species that are non-randomly replacing specialist ones [179], probably due to anthropogenic drivers such as habitat fragmentation that can more negatively affect specialists than generalists [25]. In our study, a signal that non-generalist species respond more adversely than generalist ones has been noted (Table 3D and Figure 3D).

## 5. Conclusions

The study provided an updating of long-term populations trends (1992–2019) for 76 common breeding birds in Lombardy using a statistical approach that allowed taking into account environmental bias, overdispersion and zero-inflation. Zero-inflated models largely resulted the best modelling choice in order to predict yearly population indices, and we encourage a broader application in population trend analysis than their actually employment. Overall, the bird conservation status in the region highlighted a favorable situation, with roughly three-quarters of the species showing positive or non-significant trends over the period. However, among the remnant species, 12 showed a decline greater than 50%, and 6 experienced a worrying population reduction greater than 80%, with the extreme drop of the Eurasian Skylark population (−99.65%). Identifying the causes of the observed trends, especially when dramatically negative, is the foundation of any management plan and conservation action. In this context, a functional approach linking the species traits with the population trends allowed us to identify the most threatened groups. We found a decrease of migrant birds, species with high annual fecundity and short incubation period. Plant-eaters, species feeding on invertebrates and ground-nesters also decreased. Farmland birds showed a negative trend, while woodland species resulted on increase. A weaker signal was found in relation to the species’ mobility and the degree of specialization. A trait-based approach should represent the starting point from which implement studies that test specific hypotheses aimed at identifying the direct causes responsible of the observed population dynamics. Further studies should focus on investigation of the relationship between long-term trends and species traits at large spatial scales, and on quantifying the effects of specific drivers across multiple species sharing similar functional characteristics. 

## Figures and Tables

**Figure 1 animals-11-03426-f001:**
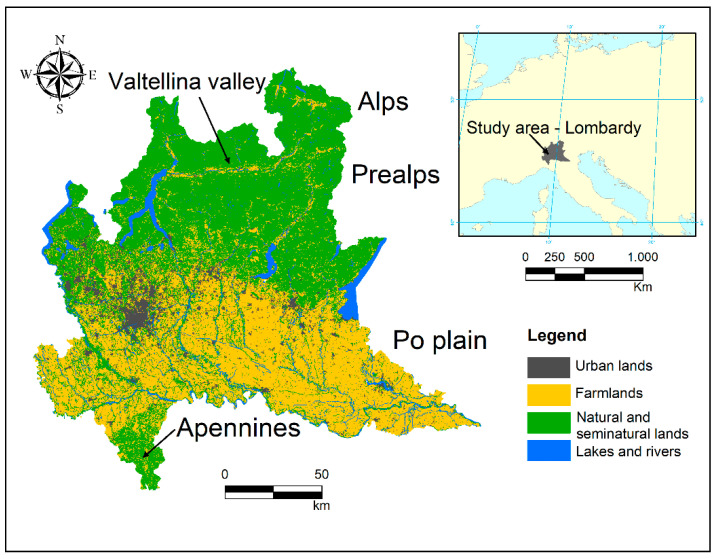
Study area. Land use refers to DUSAF 2018 digital map, level 1 of classification [55].

**Figure 2 animals-11-03426-f002:**
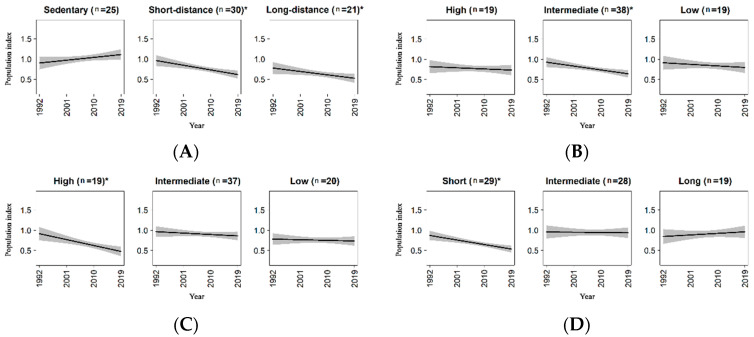
Population trends for bird species grouped according to the life history traits. (**A**) Migration strategy; (**B**) dispersal ratio; (**C**) annual fecundity; (**D**) incubation period. Y-axis represents the population index of the species (see Section 2.4.2 for details). Shaded area represents the 95% confidence interval of the regression line. The number of species in each group is shown in parentheses. The asterisk indicates the significance of the trend in relation to non-zero slope.

**Figure 3 animals-11-03426-f003:**
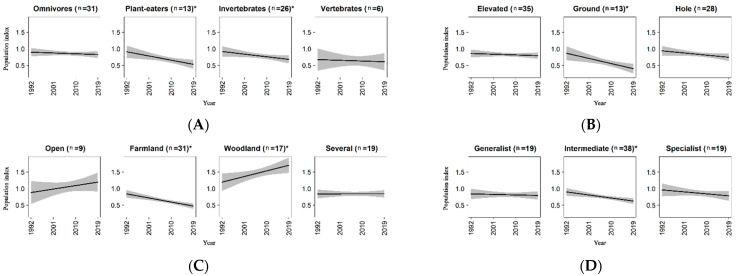
Population trends for bird species grouped according to ecological traits. (**A**) Diet; (**B**) nest type; (**C**) landscape type; (**D**) overall specialization index. Y-axis represents the population index of the species (see Section 2.4.2 for details). The number of species in each group is shown in parentheses. The asterisk indicates the significance of the trend in relation to non-zero slope.

**Table 1 animals-11-03426-t001:** Summary of weighted least square linear regression (WLS) for each species. Column “WLS model” indicates the best model selected by AIC, used to perform the linear regression. C, model with covariates dealing with environmental bias; ZIP, zero-inflated Poisson; ZINB, zero-inflated negative binomial; P, Poisson; NB, negative binomial; GAM, generalized additive model. β, estimate of regression coefficient for the explanatory variable “Year”; SE, standard error of β; t-value, t statistic; T% 1992–2019, percentage of change in the population dimension from 1992 to 2019 according to the WLS model (significant trends are marked in bold); Adj-R^2^: adjusted R-square (in case of negative value it was round to zero). Estimates and standard errors for the intercept are not showed. The common names of the species are presented according to the nomenclature suggested by [96].

Species	WLS Model	β ^1^	SE ^1^	t Value	*p* Value	T% 1992–2019	Adj-R^2^
Great Cormorant (*Phalacrocorax carbo*)	C-NB-GAM	0.050	0.007	−7.046	<0.001	**+12,060**	0.78
Black-crowned Night Heron (*Nycticorax nycticorax*)	C-NB-GAM	−0.112	0.068	−1.661	0.111	−47.86	0.07
Little Egret (*Egretta garzetta*)	C-NB-GAM	0.366	0.134	2.745	0.012	**+100.59**	0.22
Grey Heron (*Ardea cinerea*)	C-NB-GAM	0.022	0.008	2.719	0.013	**+54.76**	0.22
Mallard (*Anas platyrhynchos*)	C-ZINB-GAM	0.019	0.003	6.776	<0.001	**+525.19**	0.66
Black Kite (*Milvus migrans*)	C-ZINB-GAM	−0.053	0.022	−2.459	0.022	**−54.96**	0.18
Common Buzzard (*Buteo buteo*)	C-P-GAM	0.074	0.052	1.416	0.171	+50.42	0.04
Common Kestrel (*Falcon tinnunculus*)	C-P-GAM	0.083	0.018	4.760	<0.001	**+239.37**	0.49
Common Quail (*Coturnix coturnix*)	C-ZIP-GAM	−0.044	0.030	−1.471	0.155	**−46.83**	0.05
Common Pheasant (*Phasianus colchicus*)	C-NB-GAM	0.121	0.016	7.596	<0.001	**+591.31**	0.71
Common Moohren (*Gallinula chloropus*)	C-NB-GAM	−0.008	0.009	−0.819	0.421	−13.83	0
Northern Lapwing (*Vanellus vanellus*)	C-NB-GAM	0.218	0.033	6.514	<0.001	**+1653.42**	0.65
Feral Pigeon (*Columba livia domestica*)	C-ZINB-GAM	0.062	0.079	0.779	0.444	+30.28	0
Common Wood Pigeon (*Columba palumbus*)	C-ZIP-GAM	1.213	0.175	6.945	<0.001	**+1727.33**	0.67
Eurasian Collared Dove (*Streptopelia decaocto*)	C-ZINB-GAM	0.415	0.060	6.876	<0.001	**+196.25**	0.67
European Turtle Dove (*Streptopelia turtur*)	C-ZIP-GAM	0.253	0.279	0.907	0.374	+22.53	0
Common Cuckoo (*Cuculus canorus*)	C-ZIP-GAM	0.104	0.227	0.458	0.651	+5.81	0
Common Swift (*Apus apus*)	C-ZINB-GAM	−1.857	0.583	−3.187	0.004	**−50.01**	0.29
European Bee-eater (*Merops apiaster*)	C-NB-GAM	0.287	0.045	6.367	<0.001	**+1325.78**	0.63
Eurasian Wryneck (*Jinx torquilla*)	C-ZIP-GAM	−0.484	0.074	−6.557	<0.001	**−81.59**	0.65
European Green Woodpecker (*Picus viridis*)	C-P-GAM	0.316	0.043	7.314	<0.001	**+418.48**	0.70
Great Spotted Woodpecker (*Dendrocopos major*)	C-P-GAM	0.425	0.034	12.448	<0.001	**+1530.25**	0.87
Eurasian Skylark (*Alauda arvensis*)	C-ZIP-GAM	−0.744	0.087	−8.521	<0.001	**−99.65**	0.76
Eurasian Crag Martin (*Ptyonoprogne rupestris*)	C-NB-GAM	0.078	0.052	1.498	0.148	+64.74	0.05
Barn Swallow (*Hirundo rustica*)	C-ZINB-GAM	−1.788	0.288	−6.209	<0.001	**−67.41**	0.62
Common House Martin (*Delichon urbicum*)	C-ZINB-GAM	−0.864	0.276	−3.127	0.005	**−45.55**	0.28
Tree Pipit (*Anthus trivialis*)	C-ZIP-GAM	0.134	0.176	0.761	0.455	+14.41	0
Water Pipit (*Anthus spinoletta*)	C-ZINB-GAM	0.019	0.025	0.746	0.463	+15.12	0
Western Yellow Wagtail (*Motacilla flava*)	C-ZIP-GAM	−1.057	0.149	−7.073	<0.001	**−64.79**	0.68
Grey Wagtail (*Motacilla cinerea*)	C-P-GAM	0.003	0.034	0.088	0.931	+1.36	0
White Wagtail (*Motacilla alba*)	C-P-GAM	−0.338	0.126	−2.693	0.013	**−33.56**	0.21
Eurasian Wren (*Troglodytes troglodytes*)	C-ZIP-GAM	−0.054	0.087	−0.626	0.538	−8.91	0
Dunnock (*Prunella modularis*)	C-P-GAM	0.597	0.101	5.923	<0.001	**+309.03**	0.60
European Robin (*Erithacus rubecula*)	C-ZIP-GAM	0.140	0.053	2.640	0.015	**+35.51**	0.21
Common Nigthingale (*Luscinia megarhynchos*)	C-ZIP-GAM	−0.356	0.116	−3.055	0.006	**−35.32**	0.27
Black Redstart (*Phoenicurus ochruros*)	C-ZIP-GAM	0.161	0.021	7.523	<0.001	**+135.77**	0.71
Common Redstart (*Phoenicurus Phoenicurus*)	C-ZIP-GAM	0.569	0.121	4.682	<0.001	**+126.60**	0.48
African Stonechat (*Saxicola torquatus*)	C-ZIP-GAM	−0.687	0.143	−4.793	<0.001	**−87.83**	0.49
Northern Wheatear (*Oenanthe oenanthe*)	C-P-GAM	−0.001	0.006	−0.136	0.893	−3.67	0
Common Blackbird (*Turdus merula*)	C-ZIP-GAM	1.291	0.682	1.894	0.072	+25.15	0.10
Song Thrush (*Turdus philomelos*)	C-ZIP-GAM	0.561	0.051	11.026	<0.001	**+2869.64**	0.84
Mistle Trush (*Turdus viscivorus*)	C-ZIP-GAM	0.439	0.078	5.608	<0.001	**+468.08**	0.57
Cetti’s Warbler (*Cettia cetti*)	C-ZIP-GAM	−0.096	0.031	−3.049	0.006	**−62.35**	0.27
Melodius Warbler (*Hippolais polyglotta*)	C-ZIP-GAM	0.273	0.044	6.154	<0.001	**+346.07**	0.62
Lesser Whitethroat (*Curruca curruca*)	C-ZIP-GAM	0.141	0.042	3.320	0.003	**+153.05**	0.30
Eurasian Blackcap (*Sylvia atricapilla*)	C-ZIP-GAM	0.848	0.387	2.192	0.039	**+14.50**	0.14
Western Bonelli’s Warbler (*Phylloscopus bonelli*)	C-ZINB-GAM	0.355	0.110	3.214	0.004	**+79.62**	0.29
Common Chiffchaff (*Phylloscopus collybita*)	C-ZIP-GAM	−0.154	0.053	−2.919	0.008	**−40.64**	0.25
Goldcrest (*Regulus regulus*)	C-ZIP-GAM	−0.102	0.047	−2.184	0.040	**−41.42**	0.14
Common Firecrest (*Regulus ignicapilla*)	C-ZIP-GAM	0.363	0.084	4.349	<0.001	**+309.79**	0.44
Spotted Flycatcher (*Muscicapa striata*)	C-P-GAM	1.477	0.209	7.080	<0.001	**+490.70**	0.68
Long-tailed Tit (*Aegithalos caudatus*)	C-P-GAM	0.186	0.049	3.781	0.001	**+114.79**	0.37
Marsh Tit (*Poecile palustris*)	C-ZIP-GAM	0.295	0.049	6.006	<0.001	**+340.76**	0.60
Willow Tit (*Poecile montanus*)	C-ZIP-GAM	0.155	0.068	2.289	0.032	**+116.63**	0.16
European Crested Tit (*Lophophanes cristatus*)	C-ZIP-GAM	0.072	0.020	3.640	0.001	**+141.11**	0.35
Coal Tit (*Periparus ater*)	C-ZIP-GAM	0.046	0.040	1.155	0.261	+20.66	0.01
Eurasian Blue Tit (*Cyanistes caeruleus*	C-ZIP-GAM	0.179	0.071	2.538	0.019	**+48.78**	0.19
Great Tit (*Parus major*)	C-ZIP-GAM	1.615	0.248	6.502	<0.001	**+104.37**	0.64
Eurasian Nuthatch (*Sitta europea*)	C-ZIP-GAM	0.015	0.009	1.788	0.088	+81.84	0.09
Short-toed Treecreeper (*Certhia brachydactyla*)	C-ZIP-GAM	0.160	0.022	7.155	<0.001	**+998.50**	0.70
Eurasian Golden Oriole (*Oriolus oriolus*)	C-ZIP-GAM	0.229	0.067	3.404	0.003	**+75.75**	0.32
Red-backed Shrike (*Lanius collurio*)	C-P-GAM	−0.576	0.112	−5.128	<0.001	**−80.13**	0.52
Eurasian Jay (*Garrulus glandarius*)	C-P-GAM	0.146	0.020	7.218	<0.001	**+174.93**	0.69
Eurasian Magpie (*Pica pica*)	C-ZIP-GAM	0.279	0.029	9.541	<0.001	**+753.86**	0.80
Carrion Crow (*Corvus corone*)	C-ZIP-GAM	−0.008	0.044	−0.192	0.850	−6.24	0
Hooded Crow (*Corvus cornix*)	C-ZINB-GAM	0.395	0.301	1.309	0.204	+13.31	0.03
Common Starling (*Sturnus vulgaris*)	C-ZINB-GAM	−0.261	0.177	−1.473	0.155	−18.55	0.05
Italian Sparrow (*Passer italiae*)	C-ZIP-GAM	−1.902	0.235	−8.077	<0.001	**−71.06**	0.74
Eurasian Tree Sparrow (*Passer montanus*)	C-ZIP-GAM	−0.272	0.064	−4.253	<0.001	**−41.31**	0.43
Common Chaffinch (*Fringilla coelebs*)	C-ZIP-GAM	−0.333	0.286	−1.161	0.258	−5.45	0.02
European Serin (*Serinus serinus*)	C-ZIP-GAM	−0.111	0.345	−0.322	0.751	−4.49	0
European Greenfinch (*Chloris chloris*)	C-ZINB-GAM	−2.590	0.338	−7.657	<0.001	**−82.00**	0.72
European Goldfinch (*Carduelis carduelis*)	C-ZINB-GAM	−2.477	0.297	−8.329	<0.001	**−86.89**	0.75
Common Linnet (*Linaria cannabina*)	C-ZINB-GAM	0.034	0.044	0.767	0.452	+20.60	0
Common Redpoll (*Acanthis flammea*)	C-ZINB-GAM	−0.427	0.247	−1.727	0.098	−42.55	0.08
Eurasian Bullfinch (*Pyrrhula pyrrhula*)	C-ZIP-GAM	0.099	0.063	1.561	0.133	+73.22	0.06

^1^ Values are raised to 10^2^ to ease comparisons.

**Table 2 animals-11-03426-t002:** Weighted linear regression models showing the response of avian functional groups in relation to life-history traits. In “model output” are shown model statistics (estimate, estimated coefficient; SE, standard error; t value, t-statistic) and the significance of each coefficient (*p* value). The term “Intercept” and “Year” represent the reference group. The number of species included in each group is shown in parentheses. In “Additional tests” are shown tests of equivalence to zero of interaction effect of non-reference levels and tests of equivalence between the interaction terms for the other levels (F value: F statistic). Significant *p* values (≤0.05) for the interaction terms are marked in bold (A) Migration strategy. Groups: sedentary (reference level), short distance migrants (SDM), long distance migrants (LDM); adj-R^2^= 0.05. (B) Dispersal ratio: Groups: high dispersal ratio (reference level), intermediate dispersal ratio (IDR), limited dispersal ratio (LDR); adj-R^2^ = 0.01. (C) Annual fecundity. Groups: high annual fecundity (reference level), intermediate annual fecundity (IAF), low annual fecundity (LAF); adj-R^2^ = 0.02. (D) Incubation period. Groups: short incubation period (reference level), intermediate incubation period (IIP), long incubation period (LIP); adj-R^2^ = 0.04.

(A) Migration Strategy				
Model output				
Term	Estimate	SE	t Value	*p* Value
Intercept (25)	−14.441	9.176	−1.574	0.116
SDM (30)	40.904	11.808	3.464	<0.001
LDM (21)	33.821	12.351	2.738	0.006
Year	0.008	0.005	1.683	0.093
Year: SDM	−0.021	0.006	−3.486	**<0.001**
Year: LDM	−0.017	0.006	−2.769	**0.** **006**
Additional tests				
Null hypothesis			F Value	*p* Value
Year: SDM = 0			11.966	**<0.001**
Year: LDM = 0			5.143	**0.** **023**
Year: SDM = Year: LDM			0.391	0.532
(B) Dispersal ratio				
Model output				
Term	Estimate	SE	t Value	*p* Value
Intercept (19)	7.039	9.335	0.754	0.451
IDR (38)	15.051	11.582	1.300	0.194
LDR (19)	2.484	13.642	0.182	0.856
Year	−0.003	0.005	−0.672	0.502
Year: IDR	−0.008	0.006	−1.300	0.194
Year: LDR	−0.001	0.007	−0.176	0.860
Additional tests				
Null hypothesis			F Value	*p* Value
Year: IDR = 0			9.683	**0.002**
Year: LDR = 0			0.761	0.383
Year: IDR= Year: LDR			1.097	0.295
(C) Annual fecundity				
Model output				
Term	Estimate	SE	t Value	*p* Value
Intercept (19)	33.284	9.141	3.641	<0.001
IAF (37)	−24.554	11.909	−2.062	0.039
LAF (20)	−29.050	12.367	−2.349	0.019
Year	−0.016	0.005	−3.570	**<0.001**
Year: IAF	0.012	0.006	2.083	**0.037**
Year: LAF	0.015	0.006	2.357	**0.019**
Additional tests				
Null hypothesis			F Value	*p* Value
Year: IAF = 0			1.051	0.305
Year: LAF = 0			0.174	0.676
Year: IAF = Year: LAF			0.148	0.701
(D) Incubation period				
Model output				
Term	Estimate	SE	t Value	*p* Value
Intercept (29)	25.684	6.411	4.006	<0.001
IIP (28)	−23.327	11.436	−2.040	0.042
LIP (19)	−33.426	12.549	−2.664	0.008
Year	−0.012	0.003	−3.902	**<0.001**
Year: IIP	0.012	0.006	2.063	**0.039**
Year: LIP	0.017	0.006	2.682	**0.007**
Additional tests				
Null hypothesis			F Value	*p* Value
Year: IIP = 0			0.022	0.881
Year: LIP = 0			0.643	0.423
Year: IIP= Year:LIP			0.492	0.483

**Table 3 animals-11-03426-t003:** Weighted linear regression models showing the response of avian functional groups in relation to ecological traits. In “model output” are shown model statistics (Estimate, estimated coefficient; SE, standard error; t value, t-statistic) and the significance of each coefficient (*p* value). The term “Intercept” and “Year” represent the reference group. The number of species included in each group is shown in parentheses. In “Additional tests” are shown tests of equivalence to zero of interaction effect of non-reference levels and tests of equivalence between the interaction terms for the other levels (F value: F statistic). Significant *p* values (≤0.05) for the interaction terms are marked in bold. (A) Diet. Groups: omnivores (references level), plant-eaters (PLA), invertebrates (INV), vertebrates (VER); adj-R^2^ = 0.01. (B) Nest type. Groups: elevated-nesters (reference level), ground-nesters (GR), hole-nesters (HL); adj-R^2^ = 0.02. (C) Landscape type. Groups: open natural habitat (reference level), farmland (FAR), woodland (WOO), several (SEV); adj-R^2^ = 0.11. (D) SI. Groups: generalist species (reference level), intermediate species (INT), specialist species (SPE); adj-R^2^ = 0.02.

(A) Diet				
Model output				
Term	Estimate	SE	t Value	*p* Value
Intercept (31)	6.242	7.250	0.861	0.389
PLA (13)	22.686	12.764	1.777	0.076
INV (26)	12.264	11.541	1.063	0.288
VER (6)	−0.662	20.755	−0.032	0.975
Year	−0.003	0.004	−0.743	0.458
Year: PLA	−0.011	0.006	−1.791	0.074
Year: INV	−0.006	0.006	−1.069	0.285
Year: VER	0.000	0.010	0.021	0.983
Additional tests				
Null hypothesis			F Value	*p* Value
Year: PLA = 0			7.232	**0.007**
Year: INV = 0			3.898	**0.048**
Year: VER = 0			0.065	0.799
Year: PLA = Year:INV			0.579	0.446
Year: PLA = Year: VER			1.111	0.292
Year: INV = Year: VER			0.356	0.551
(B) Nest type				
Model output				
Term	Estimate	SE	t Value	*p* Value
Intercept (35)	5.627	6.834	0.823	0.410
GR (13)	29.582	13.532	2.186	0.029
HL (28)	9.696	10.731	0.904	0.366
Year	−0.002	0.003	−0.703	0.482
Year: GR	−0.015	0.007	−2.203	**0.028**
Year: HL	−0.005	0.005	−0.903	0.367
Additional tests				
Null hypothesis			F Value	*p* Value
Year: GR = 0			8.790	**0.003**
Year: HOLE = 0			3.070	0.080
Year: GR = Year: HL			1.976	0.160
(C) Landscape type				
Model output				
Term	Estimate	SE	t Value	*p* Value
Intercept (9)	−22.340	20.764	−1.076	0.282
FAR (31)	50.341	21.718	2.318	0.021
WOO (17)	−14.078	26.297	−0.535	0.593
SEV (19)	22.952	22.164	1.036	0.301
Year	0.012	0.010	1.127	0.260
Year: FAR	−0.025	0.011	−2.338	**0.020**
Year: WOO	0.007	0.013	0.551	0.581
Year: SEV	−0.012	0.011	−1.045	0.296
Additional tests				
Null hypothesis			F Value	*p* Value
Year: FAR = 0			18.505	**<0.001**
Year: WOO = 0			5.513	**0.019**
Year: SEV = 0			<0.001	0.977
Year: FAR = Year: WOO			14.153	**<0.001**
Year: FAR = Year: SEV			7.572	**0.006**
Year: WOO = Year: SEV			4.426	**0.036**
(D) Overall specialization index				
Model output				
Term	Estimate	SE	t Value	*p* Value
Intercept (19)	4.256	9.046	0.471	0.638
INT (38)	16.896	11.230	1.505	0.133
SPE (19)	9.934	14.312	0.694	0.488
Year	−0.002	0.005	−0.381	0.704
Year: INT	−0.008	0.006	−1.511	0.131
Year: SPE	−0.005	0.007	−0.691	0.490
Additional tests				
Null hypothesis			F Value	*p* Value
Year: INT = 0			9.411	**0.002**
Year: SPE = 0			1.446	0.230
Year: INT = Year: SPE			0.300	0.584

## Data Availability

Full dataset is available at the Department of Earth and Environmental Sciences, University of Milan-Bicocca, Piazza della Scienza 1, 20126, Milan, Italy, Building U1.

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
