# Peer review of "Species Traits Drive Long-Term Population Trends of Common Breeding Birds in Northern Italy"

_animals, 2021, doi:10.3390/ani11123426_

Round 1

Reviewer 1 Report

Article animals-1437716

Tirozzi et al. Analyze in " Species traits drive long-term population trends of common breeding birds in Northern Italy" long term populations and try to assess trends observed in functional groups. The strength of the paper as presented is the data baseline which is very good and extremely valuable and the idea to analyze the functional traits also is certainly a strong asset. However, I found it difficult to follow all the diverse analyses and why you performed what to show the trends. The simplest way to give a trendline per species would be to draw your observations per species per year (and maybe per study?) to show the reader that is the data you have. Then I would go on to adjust for occurrence and detectability bias – a part that to my opinion is missing form your study, which in turn would increase the comparability of the sites and in-between the 5 studies. Instead of Table 1 and 2 you should try to have a figure showing theses data as a trendline estimate – if ot done in your figures, but then it would be redundant and I would move the table 1 and 2 into supporting information. For me is not clear if the figures and tables give the same content or not hence please clarify.

The conclusion of the results (45% of the species increased, 24% decreased, and 31% non trends) and  Life history traits analyses (decrease of migrants, and of species with high annual fecundity) are interesting and ifteh models and data analysis are better presented then this is a very valuable and interesting result which must be published.

While I think the general idea and the data and some of the analysis are top, I found it difficult to follow other parts and some of the analysis. I comment this in detail in the section below. In general, the patchwork dataset of 5 projects with partially overlapping but also discrepant standardization is inflicting some issues which I think are not correctly addressed or at not described adequately:

  • Occurrence instead of abundance
  • Detectability of the species varies but seems to be not taken into account in the analysis; this might pose an issue for the correct interpretation of the results, particularly since the observation distance is unlimited hence NOT standardized.

From this arises an issue on performing many statistical and other model approaches which is most prominently visible in the model naming. The names of the models are certainly correct, but they are anything but intuitively understandable and I had a har time to understand what code means whet.

Further minor comments:

Ll75: it should be Germany instead of German

Ll136-139: there serval issues here that are probably easy to adjust:

Standard: I assume you describe this further below, but so far you need to be more specific. a standard method is not enough in your case you have to adjust for methodological differences in your models, even if the difference are minute as described.

Unlimited distance: that is not very informative and not very helpful for your models. How did you arrange then the overlap by 500 m? There is a range of species that you may have double counted (flying raptors very far away, way beyond 500 m) and how did you adjust for this potential error? How did you assess the issue of spatial overlap for some species?

500 m: did you check and if needed adjust for spatial autocorrelation? if yes, how and if not, why not?

Ll143: bird abundance: I disagree. Even if you sampled at all localities the same way, you only have occurrence per species, particularly since you have unlimited distance. This is the critical point: how did you assess the occurrence at your count localities with unlimited distance? This is not explained but is the essential point to understand your data. I think if you cannot adjust that issue you should down tune the wording form abundance to occurrence throughout the text.

In consequence you have occurrence (your relative abundance) for each of your 5 studies, but not for all studies merged, because the methodological differences prevent this?

Ll147: good, BUT how did you avoid overestimation of very good detectable birds compared to very secretive birds you just see within a few meters or do not hear them? Particularly species certainly not as far detectable as e.g. overflying raptors? There is a strong bias against some species if you did not assess occurrence and detection probability in your models. I have not seen that you did so. You can use mark4 or I think the package vegan() in R to do so (not sure about vegan, there has been an attempt to implement but it might be not working properly or the attempt failed).

Ll158: You describe here how you address some few of my previous concerns, BUT please make you reader having a standalone paper so explain what the cited references are talking about in some few relevant keywords. Also I recommend mentioning the first author of the study here (and hope it does not contradict the journal guidelines), then 90% of all your readers know what has been done.

ll162: What is environmental bias? you mean bird communities or bird species respond to environmental drivers? or do you mean methodological (detectability) bias? please specify, I have not seen this term without further detailed definitions.

Ll164: details needed on how and what? did you use estimators for the population estimates? if yes which ones? if not how did you calculate this index? Why did you use an index anyway? There might be better ways to do so, pending on your detailed standardization procedures form above.

Ll166-169: all certainly correct. BUT having read your model names and explanations is not very helpful for the rest of the text - in the results section your readers will have forgotten what model name refers to what model parameters in detail. Either please find a more intuitively model name or repeat the details in your Table 1 and Table 2 (and also in the results section) so that the readers are reminded on the context and what name means what.

Ll174 fractional cover - ? How did you do that? What criteria? How does a GAM provide you a population index?

Ll176: "environmental bias" The term is confusing, what do you mean? You should explain this in the introduction and link your work to it if this is so important. Otherwise you should explain here in detail what you refer to, see also other comments. I did not see a convincing explanation throughout the text what you refer to with "environmental bias"

Ll 210: I think most of the text could be moved in a relatively small table with the parameter name, short description and how you calculated the indices followed by the citation ##.

Ll274: you calculated how many trait-based models? is there no way to get the relative weight for the traits and then assess the relative importance? To my opinion the relative importance would be very interesting as to compare the relative strength between the taxa.

Reviewer 2 Report

I have enjoyed reading this interesting and very well-written study with much detailed information. The study has been performed with hard work and extracting a large database, including detailed information about the avian traits and environmental variables. The manuscript presents a long-term and point-count method dataset that, in ornithological studies, is rare and is very well appreciated. The main findings of this study are quite relevant in conservation and ecology, and most of the methods applied were well-addressed. Introduction and methods sections are readable and written meticulously. I provide some suggestions for the tables and figures presentation. In the current form, I found that the model tables are long and not intuitive. Also, the figures can be arranged a little bit to improve their communication style. The discussion section is the best-addressed section of the article. For these reasons, I think the Animals readers could very much appreciate this article, and I recommend its publication after a minor review. I will be very grateful to check the changes. More details in the specific section of comments to the authors.

Introduction

Page 2, line 75:  Would you mean ‘Germany’?

Results

Page 7, line 306: Please, can you add in the legend the meaning of CNBGAM, CZINBGAM, etc.?

Page 7, line 311: Please, remove additional points.

Pages 12-13, lines 327-341 (Table 2): I suggest moving this table to the supplementary material.

Page 13, Figure 2: I suggest disposing each group in one column (a plaid with three figures horizontal x 4 vertical figures). Instead of indicating the groups with letters, you can use their names (migration, dispersal, etc.). Alternatively, I suggest separating each group with lines making more evident their limits and indicating their names above with the real name instead of letters.

Pages 13-15, lines 365-379 (Table 2): I suggest moving this table to the supplementary material.

Page 16, Figure 3: I suggest separating each group with lines making more evident their limits and indicating their names above with the real name instead of letters.

Reviewer 3 Report

The authors explore variation in breeding population size for a number of species in northern Italy. They use data from a set of long-term surveys and provide detail information on how this sampling effort has been conducted over time. Their findings are well explained and discussed, they provide an useful base for further research. I have just few comments, they are mostly about including more detail in some parts that were not clear to me or that I think are necessary to fully understand the study.  

Line 13. “regarding species groups….”

Line 13-15. Please add “population” so it is clear it is a population decline

Line 16. The sentence “In conclusion, our study provided a portrait of the status of common breeding birds in the Region, pointing out which species need of major conservation attention” is quite similar to the one following it, please change it.

Line 49. What does “conspicuous” mean in this sentence?

Line 51. I would replace the word “precious” for something else

Line 56. The word “integrity” might not fit in this context.

Line 59. What trends the authors refer to?

Line 74. This sentence is difficult to follow.

Line 93. What kind of heterogeneity?

Line 104. River Po

Line 106.  Replace the word “much” for something else

Line 119. It is not clear that “ raw data” means, point counts?

Line 147 -152. A more detailed explanation is necessary to understand this part. For instance, where those figures come from?

Line 159. It is necessary that a brief description of the method is included in here, to provide a general idea at least.

Line 166. I suggest giving different names to the models. If all of them are GAM, then I would remove the “GAM” part from the name. Also, I would use “_” to link the different features of the models eg. C_ZIP.

Line 166. I think the manuscript will benefit of an explanation on the differences across models, for the readers that are unfamiliar with GAM.

Line 169. “we used the best performing”. Also, please mention how you compare model performance.

Line 180. This sentence is not completely clear to me, please improve it

Line 250. “as the original study”

Line 266. Please clarify that SI stands for specialization index

Table 1. I would like the authors to explain how the Cormorant had such a massive increase in population size with such a low beta coefficient. In general, I think an explanation is needed on how some species reach very large T% values.

Line 389. This sentence needs rewording.

Line 407. Please replace “predicted” by a more suitable word.

Line 417. “needs”

Line 428. Remove the last bit of the sentence, the part about the hypothesis.

Line 440. “Decline”

Line 582.  Remove “Anyway”

Line 588. Remove “for”

Line 589. Although the discussion is quite well done and a bit lengthy, I would still like to see some discussion on the findings of this study fit into the red list from IUCN or another international (and relevant) criteria for classifying species in threatened and non-threatened.  

Round 2

Reviewer 2 Report

Dear Authors, many thanks for all improvements made in the previous version of the manuscript. I think this manuscript is ready for final acceptance. This work represents an important addition to the avian population trends and species sharing similar ecological and life-history traits. The most relevant contributions are the large and detailed data collection through many years. Congratulations to the authors.

Author Response

We thank the Reviewer for his/her approval.
We very much appreciated his/her positive comments.

Reviewer 3 Report

I appreciate the authors have addressed my comments. 

Author Response

We thank the Reviewer for his/her approval and positive comments.